# Developing Cardio-Oncology Programs in the New Era: Beyond Ventricular Dysfunction Due to Cancer Treatments

**DOI:** 10.3390/cancers15245885

**Published:** 2023-12-18

**Authors:** Alexandra Pons-Riverola, Herminio Morillas, Javier Berdejo, Sonia Pernas, Helena Pomares, Leyre Asiain, Alberto Garay, Adela Fernandez-Ortega, Ana Carla Oliveira, Evelyn Martínez, Santiago Jiménez-Marrero, Elena Pina, Eduard Fort, Raúl Ramos, Lídia Alcoberro, Encarnación Hidalgo, Maite Antonio-Rebollo, Laia Alcober, Cristina Enjuanes Grau, Josep Comín-Colet, Pedro Moliner

**Affiliations:** 1Cardio-Oncology Unit, Bellvitge University Hospital—Catalan Institute of Oncology, L’Hospitalet de Llobregat, 08908 Barcelona, Spain; aponsr@bellvitgehospital.cat (A.P.-R.); herminiomorillas@bellvitgehospital.cat (H.M.); fjberdejo@bellvitgehospital.cat (J.B.); spernas@iconcologia.net (S.P.); hpomares@iconcologia.net (H.P.); lasiasin@iconcologia.net (L.A.); agaraym@bellvitgehospital.cat (A.G.); afernandezortega@iconcologia.net (A.F.-O.); acoliveira@iconcologia.net (A.C.O.); emperez@iconcologia.net (E.M.); sjimenezm@bellvitgehospital.cat (S.J.-M.); epina@bellvitgehospital.cat (E.P.); rramosp@bellvtgehospital.cat (R.R.); 2Cardiology Department, Bellvitge University Hospital, L’Hospitalet de Llobregat, 08907 Barcelona, Spain; lalcoberro@bellvitgehospital.cat (L.A.); ehidalgoq@bellvitgehospital.cat (E.H.); cenjuanes@bellvitgehospital.cat (C.E.G.); jcomin@bellvtigehospital.cat (J.C.-C.); 3Bio-Heart Cardiovascular Diseases Research Group, Bellvitge Biomedical Research Institute (IDIBELL), L’Hospitalet de Llobregat, 08908 Barcelona, Spain; 4Medical Oncology Department, Catalan Institute of Oncology, L’Hospitalet de Llobregat, 08908 Barcelona, Spain; 5Clinical Haematology Department, Catalan Institute of Oncology, L’Hospitalet de Llobregat, 08908 Barcelona, Spain; 6Radiation Oncology Department, Catalan Institute of Oncology, L’Hospitalet de Llobregat, 08908 Barcelona, Spain; 7Haematopoietic and Lymphoid Tumours Group, Bellvitge Biomedical Research Institute (IDIBELL), L’Hospitalet de Llobregat, 08908 Barcelona, Spain; 8Radiobiology and Cancer Group, ONCOBELL Program, Bellvitge Biomedical Research Institute (IDIBELL), L’Hospitalet de Llobregat, 08908 Barcelona, Spain; 9Centro de Investigación Biomédica en Red de Enfermedades Cardiovasculares (CIBERCV), 28029 Madrid, Spain; 10Thrombosis and Haemostasis Unit, Bellvitge University Hospital, L’Hospitalet de Llobregat, 08907 Barcelona, Spain; 11Pharmacy Department, Catalan Institute of Oncology, L’Hospitalet de Llobregat, 08908 Barcelona, Spain; efort@iconcologia.net; 12Oncogeriatrics Department, Catalan Institute of Oncology, L’Hospitalet de Llobregat, 08908 Barcelona, Spain; marebollo@iconcologia.net; 13Primary Care Service Delta del Llobregat and IDIAP, Catalan Health Service, 08007 Barcelona, Spain; lalcober.apms.ics@gencat.cat

**Keywords:** Cardio-Oncology, cardiovascular disease, cancer, cardiotoxicity, program, optimal cancer therapy

## Abstract

**Simple Summary:**

Modern cancer therapies have achieved a remarkable improvement in overall survival and patients’ quality of life. However, cardiovascular toxicities are still a major concern. A specific Cardio-Oncology unit is key to offering patients with cancer the best approaches to treatment while minimizing adverse cardiac effects. Moreover, this area of medicine requires a large expertise and has limited trials on which to base decision-making. The development of structured Cardio-Oncology programs leads to better patient care and generates scientific evidence that may impact patient’s survival outcomes. In this review, we summarize our experience and describe the essential steps to consider when creating a program.

**Abstract:**

Cardiovascular disease is a common problem in cancer patients that is becoming more widely recognized. This may be a consequence of prior cardiovascular risk factors but could also be secondary to the anticancer treatments. With the goal of offering a multidisciplinary approach to guaranteeing optimal cancer therapy and the early detection of related cardiac diseases, and in light of the recent ESC Cardio-Oncology Guideline recommendations, we developed a Cardio-Oncology unit devoted to the prevention and management of these specific complications. This document brings together important aspects to consider for the development and organization of a Cardio-Oncology program through our own experience and the current evidence.

## 1. Introduction

There is growing evidence as to the connection between cancer and cardiovascular disease (CVD) beyond the several risk factors they share like aging, obesity, diabetes, hypertension or dyslipidemia [1]. The incidence of cancer in patients with chronic heart failure (HF) is estimated at about 19 to 34 per 1000 person-years [2], which is higher than in the general population [3]. Additionally, recent studies have shown that it is one of the main causes of death among chronic HF patients [4]. Conversely, it is well known that several cancer treatments have a deleterious effect on the cardiovascular system. Some of them are known to be toxic for the myocardium and eventually lead to ventricular dysfunction and HF. The incidence of left ventricle dysfunction and HF is estimated to be between 2 and 33% in cancer patients, depending on the cancer treatment they receive and their basal characteristics [5].

Cardio-Oncology (CO) is a new and rising field that was initially created to perform an early diagnosis and treatment of cardiotoxicity (CTOX), secondary to anthracyclines or anti-human epidermal growth factor receptor 2 (HER2) agents, which enable cancer patients to withstand their treatment and therefore increase their survival rate. HF is known as one of the most critical adverse effects of cancer treatment, as it has an important impact on cancer prognosis [3]. However, as knowledge in this field grows, modern CO goes beyond solely CTOX screening and deals with a wide spectrum of cardiovascular effects related to cancer therapies. This includes arrhythmias, valvular heart disease, pericardial disease and early atherosclerosis, among others.

CO is a challenging and growing discipline that aims to provide complete CV care for cancer patients and survivors. Evidence shows that multidisciplinary programs within Oncology, Hematology and Cardiology improve clinical outcomes in cancer patients [6,7], as they guarantee cardiac optimization and cancer treatment continuation [8]. The goal of this review is to cover the current key aspects of the organization of CO units and provide practical tips to develop new CO programs, improve the health care and optimize the cardiac management of cancer patients.

## 2. Organization of a Cardio-Oncology Unit

### 2.1. Objective of the Cardio-Oncology Unit

The aim of a Cardio-Oncology Unit is to provide specialized multidisciplinary approach and consistent, continuous, coordinated and cost-effective care during the cancer process [9]. It consists of the prevention, diagnosis and treatment of cancer patients at risk of cardiotoxicity. It also involves the monitoring and treatment of cancer patients at risk or with concomitant CV diseases. The final goal of CO services is to facilitate optimal cancer treatments and prevent the unwarranted withdrawal of treatment.

Barros-Gomes et al. describe the objectives in their CO practice in the Mayo Clinic as follows: (1) to facilitate the diagnosis, monitoring and therapy of cancer treatment related cardiovascular complications; (2) to evaluate the baseline cardiovascular risks prior to cancer treatment and implement strategies for reducing the risk of developing cardiovascular complications; and (3) to assist the patient with cardiovascular care through long-term follow-up [10].

### 2.2. Components of Cardio-Oncology Team

Cardio-Oncology is a discipline that involves different specialized professionals and demands direct communication between them to discuss shared patients. Therefore, a multidisciplinary team is crucial [9,11]. The nucleus of the CO team is composed of cardiologists (usually with a special interest and experience in the management of cardiac conditions in cancer patients), medical oncologists, radiation oncologists, hematologists and specialized nurses. Apart from the core members, the CO team also needs a close relationship with other professionals like family doctors, pathologists, radiologists, the palliative care team, pharmacists, cardiac surgeons, internists, etc. With regard to the cardiology team, Clinical Cardiology and Cardiology Imaging are usually the axis of the CO team, but multiple cardiology subspecialties may also be involved. They include the inpatient cardiology ward, cardiology critical care unit, invasive cardiology unit and heart failure specialists, among others.

### 2.3. Cardio-Oncology Programs

Ideally, a Cardio-Oncology program should cover five different areas: clinical, research, training, innovation and quality indicator evaluation (Figure 1).

Cardio-Oncology is a broad and complex area of knowledge which requires a multidisciplinary approach. To be successful, a comprehensive Cardio-Oncology program should comprise different sections. The clinical part takes place mainly in the outpatient clinic, but should also cover inpatient and interprofessional consultations. Easy and fast accessibility and multidisciplinary meetings are essential for the promotion of shared decision-making. As lack of evidence is frequent in some fields, creation of institutional databases, engaging in national and international societies and promoting participation in clinical trials and multicentric studies constitute important steps within the research section. An appropriate training is achieved through educational sessions, fellowship and mentorship programs. Lastly, Cardio-Oncology activity and results should be measured by thorough quality indicators.

#### 2.3.1. Clinical Program

The clinical program should include outpatients clinics, in-patients consultation services and interprofessional consults [9].

The CO outpatient clinic is the main activity in a CO program. Most CO consultations should be organized as a face-to-face day-case model, which reduces the number of visits to the hospital and avoids treatment delay. The aim of a day-case model is to provide clinical assessment, non-invasive investigations (blood tests with cardiac biomarkers, electrocardiogram and echocardiography) and multidisciplinary discussion on the same day. Invasive cardiac investigations, if appropriate, can be delivered in collaboration with the cardiology department (advanced cardiac imaging and interventional procedures). The virtual outpatient clinic, using a telephone call or videoconferencing, is an option to connect with patients without in person appointment, e.g., to monitor vital signs or to share test results. A CO program can be established taking advantage of existing resources, such as a specialized nursing staff and a heart failure program, thereby patients who develop significant cardiotoxicity could be transferred to HF Unit for a more specific management [12].

As for the inpatient consultation services, the CO service should also provide advice relative to oncology and hematology inpatients who develop new cardiac symptoms, as well as the cardiology inpatient with cancer. The inpatient consultation service is only possible in large CO services with more than one practicing cardio-oncologist, and ideally should be a same or next day review [13]. The CO team will also oversee organizing the patient follow-up at discharge. In some tertiary hospitals with an important number of cancer inpatients, an on-call cardiologist from the CO team should be considered.

Regarding interprofessional consults, electronic methods (e-consults) can be used as a fast and efficient way of communication between different CO specialists or with other doctors (general cardiologists, family doctors, pharmacists, etc.), particularly with regard to referral for CO consultation or recommendations and concrete advice about patient care. This method enables cardiologists to further assist oncologists and hematologists in assessing risk factors and managing existing cardiovascular disease without necessitating direct contact with the patient [10]. Polypharmacy is very frequent in elderly patients and can lead to drug–drug interaction. Interprofessional consults with pharmacists are fundamental and can reduce potentially severe adverse drug-related events. Ideally, interprofessional consults should be open access and guarantee consultative advice within 24 h, to avoid any delay in treatment initiation [12].

Multidisciplinary team meetings are essential to facilitate shared decision-making around complex patients, as they lead to face-to-face discussions between cardiologists, oncologists and hematologists about continuation, as well as the modification or interruption of a specific cancer treatment. In a multidisciplinary team, it is imperative to have a shared electronic medical record that allows for the easy and efficient exchange of information. Both cardiology and oncology medication should also be updated in the medical history [12].

CO service organization can be quite challenging, as patients often need to be seen within a week to avoid any treatment delay. The creation of protocols for referral and specific clinical pathways adapted to the available infrastructure is mandatory to ensure that efforts and resources are dedicated to the patients who can benefit most from them. Flexibility and a self-management model in the scheduling of patient appointments is key to providing assistance in a timely fashion and to adapt to the dynamic nature of CO patients and their needs [12,14].

#### 2.3.2. Research Program

Cardio-Oncology is a recently created and evolving subspecialty with limited trials on which to base decision-making. Thus, a great deal of the management of the patients relies on consensus or expert opinion. One of the main goals of creating a CO program is to participate in developing high-quality scientific evidence.

A CO service should collect clinical data of patients attended in the outpatient clinic. Advanced and multidisciplinary imaging and CO biobanks are also essential to build new prediction models of cardiotoxicity. In addition to focusing in a concrete area of local research and expertise, it is crucial to participate in multicentric studies and clinical trials. Being an active member of national and international societies is key to becoming involved in collaborative research projects [14].

#### 2.3.3. Training Program

Cardio-Oncology training should have an structured program that includes organized educational sessions with the Cardiology, Hematology and Oncology teams [14]. Like any other subspecialty, training medical students, cardiology residents and fellows in Cardio-Oncology is fundamental. It would also be worthwhile to regulate a fellowship in Cardio-Oncology and create mentorship programs for cardiologists interested in this field. This would help increase the visibility of CO and its program among other health care professionals and institutions.

Establishing educational opportunities to provide additional experience in CO, like professional seminars and conferences, is also recommended for all CO team members.

Education and support for patients also needs to be considered, so they can learn more about their disease and treatment [12].

#### 2.3.4. Innovation in Cardio-Oncology

Despite recent advances in Cardio-Oncology and the increase in information and evidence-based approaches in the diagnosis, management and treatment of oncological patients with cardiovascular disease, there are still many gaps in knowledge and areas of uncertainty.

Collaborative groups may share information and databases that increase awareness of these kind of patients and improve outcomes. Sharing experience and knowledge may create a bigger database that will make it possible to get specific and copious information and data that can extrapolated to daily clinical practice.

Artificial intelligence, patient-centered data, tele-medicine and remote follow-up are several of many fields that need to be explored and exploited in the next future.

Finally, individualized medicine derived from targeted treatment in specific scenarios may lead to treatments with greater efficacy and fewer secondary effects [15].

#### 2.3.5. Quality Indicators

There is increasing interest in discovering new tools that permit a comprehensive evaluation of quality of care, including structural and process indicators and outcomes in cardiovascular disease. Geographical and social variation in medical care delivery, as well as the difficulty in the assessment of outcomes and the need to invest in closing the so called “evidence-practice gap”, has led several medical societies to try to unify and establish shared pathway goals in the management and outcomes of patients with cardiovascular disease.

Currently, the use of quality indicators (QI) to evaluate medical practice is well accepted, as they may serve as a way to promote and enhance evidence-based medicine, through quality improvement, benchmarking of care providers and accountability. Specific pathways have been proposed to establish reliable QI. The most acknowledged program divides the QI development process into four steps: identifying the domain of care, constructing candidate quality indicators, selecting the final quality set and assessing feasibility [16].

A consensus document from the field of Cardio-Oncology that summarizes the QI has recently been published. The main domains emphasized are the structural framework, baseline cardiovascular risk assessment, cancer treatment related cardiovascular toxicity, predictors of outcomes and the monitoring of cardiovascular complications during cancer therapy [17,18].

### 2.4. Pathway of Care

Cardio-Oncology consultations are aimed at patients at considerable risk of CV complications related to anticancer treatment. Lancellotti P et al., in a report from the ESC Cardio-Oncology Council, define high risk patients as: (1) patients receiving potentially cardiotoxic treatment; (2) patients prior to cancer surgery if they have previous CV disease or are expected to receive additional cancer treatment; (3) patients who develop CV symptoms during oncological treatment; (4) patients receiving cancer treatment who develop asymptomatic newly reduced cardiac function; (5) patients with prior childhood cancer treatment; (6) those planning pregnancy or those who develop CV symptoms during pregnancy [9].

The new Cardio-Oncology guidelines also provide tools to stratify cardiovascular toxicity risk of many cancer treatments [17]. Low-risk patients can follow regular oncology monitoring without special cardiology follow-up. The main pathway of care in CO programs is the CV assessment and management of cancer patients before, during and after the cancer therapy (Figure 2).

The patient journey starts before receiving cancer treatment, where a comprehensive evaluation of cardiovascular risk should be carried out. High- and very high-risk patients may benefit from primary prevention strategies. Once cancer therapy begins, proactive monitoring and early detection of cardiac toxicity is warranted. When needed, cardiovascular treatment should be prescribed as soon as possible, in order to minimize cancer therapy interruptions. Finally, patients who successfully finish their cancer treatment should be included in a long-term survivorship program, which can be done in collaboration with general practitioners.

The aim of cardiac monitoring during cancer treatment is the early detection of cardiovascular complications. It is not only cardiotoxicity related to left ventricle dysfunction and heart failure, but also arrhythmias, hypertension, vascular toxicity, ischemia, valvular heart and pericardial disease, among others [10]. The main instruments are electrocardiograms, blood tests with cardiac biomarkers (troponin and natriuretic peptides) and strain imaging echocardiography. Timing of the cardiac monitoring depends on the anticancer treatment and the patient risk profile [19]. Advanced cardiac imaging or invasive testing might also be necessary in some patients [20].

As previously indicated, in addition to monitoring any ongoing cancer treatment, it is also crucial to have long-term surveillance programs for cancer survivors to provide appropriate follow-up and prevent the development of late-onset CV disease, as HF may develop several years after cancer treatment [17]. Conversely, cancer therapy, including radiotherapy, increases CV risk and the risk of developing coronary artery disease in the future. Communication and coordination with general practitioners and general cardiologists will be imperative for a proper follow-up.

Another important consideration is the standardization of clinical processes and care [10]. Local clinical protocols and referral pathways are basic in the Cardio-Oncology practice to reduce interindividual variability and minimize delays.

Appendix A shows the clinical care pathway and the main components of a Cardio-Oncology program.

## 3. Disease-Specific Clinical Pathways

### 3.1. Cardiotoxicity and Left Ventricle Dysfunction

Recent studies have found a high incidence (37.5%) of patients with deteriorating ventricular function during certain types of chemotherapy. Even though only severe cardiotoxicity (defined by asymptomatic LV ejection fraction < 40% or HF) has been strongly related to all-cause mortality, milder cardiotoxicity forms should represent a warning to consider a closer follow-up [21].

Before the beginning of any potentially cardiotoxic cancer treatment, patients should be studied to assess their cardiotoxicity risk (low, medium, high or very high) and cardiovascular diseases. Reviewing previous medical history, a complete physical examination, an electrocardiogram, troponin determination and strain imaging echocardiography are recommended for a complete risk evaluation [5]. Using clinical scales like the HFA-ICOS risk assessment tool is also advisable [17,22]. It is crucial to detect subclinical cardiac abnormalities which may influence clinical decisions. Baseline cardiovascular risk and the specific cancer treatment will allow both oncologists and cardiologists to decide the best treatment approach and the intensity of monitoring and medical follow-up during cancer therapy for each patient [20].

The 2022 ESC Guidelines recommend that cancer patients at high or very high risk for cardiotoxicity undergo a comprehensive CV evaluation before scheduled anticancer therapy, preferably by a Cardio-Oncology specialist [17]. High/very high-risk patients are defined by the sum of some CV risk factors, previous cardiotoxicity or exposure to cardiotoxic agents. A background of HF, a left ventricle ejection fraction (LVEF) <50% or cardiomyopathy also characterize patients as high-risk or very high-risk [14]. Elevated levels of natriuretic peptides or troponin at baseline are some criteria of medium risk.

During potential cardiotoxic therapies, such as anthracyclines, medium-, high- and very high-risk patients should have increased surveillance with periodical clinical assessment, electrocardiography (ECG), cardiac biomarkers and left ventricle (LV) function monitoring through echocardiography.

Cardiac biomarkers like troponin and natriuretic peptides are markers of initial cardiac injury and may predict LV systolic dysfunction and eventual HF development [23]. Echocardiography is the method of choice for the detection of myocardial dysfunction before, during and after cancer therapy. Global longitudinal strain has been shown to detect cardiac dysfunction at an early stage [24,25]. A 15% or more absolute reduction in GLS from baseline may suggest a risk of cardiotoxicity [20]. Other cardiac imaging techniques like nuclear cardiac imaging (MUGA) and cardiac magnetic resonance may be appropriate in some patients.

When cardiotoxicity is detected, chemotherapy should be reconsidered and specific cardiac treatment with angiotensin-converting enzyme inhibitors and beta-blockers should be started to avoid further LV dysfunction and the progression to clinical HF [17,20]. Guideline-based HF therapy is recommended in both patients with symptomatic HF and asymptomatic cardiac dysfunction [17].

From the oncological perspective, dexrazoxane and the use of liposomal anthracyclines instead of conventional ones have shown to reduce the future risk of cardiac dysfunction by approximately two-thirds in patients who have already received high cumulative anthracycline doses [26,27]. Recently, high-dose atorvastatin proved to decrease the proportion of patients with a significant decline in left ventricular ejection fraction when receiving large doses of anthracycline-based chemotherapy [28].

### 3.2. Coronary Artery Disease

It is known that chemotherapy (specially fluoropyrimidines such as 5-fluorouracil or capecitabine, cisplatin and some immune and targeted therapies) can cause coronary artery disease (CAD) with either a direct vasospastic effect or endothelial injury causing acute arterial thrombosis [29]. Furthermore, previous mediastinal radiotherapy may accelerate drug-related coronary damage [17,30].

The identification of patients with CAD should be done before initiating cancer treatment with clinical assessment based on medical history, age, and gender. A few data suggest that pre-existing CAD considerably increases the risk of developing treatment-related CAD [17].

Clinical evaluation and non-invasive assessment of myocardial ischemia are recommended in cancer patients with angina [31]. In some patients with persistent symptoms, invasive management with a coronarography and revascularization can be appropriate. Short regimens of dual antiplatelet therapy should be adopted in the majority of cases if there is no high ischemic risk [32]. However, some patients can be thrombocytopenic during chemotherapy. That represents a particular challenge and will need multidisciplinary management. In addition, bleeding risk secondary to antiplatelet and anticoagulant drugs is generally increased in cancer patients, so prophylactic antiplatelet treatment is usually not recommended.

In the particular case of patients treated with pyrimidine analogues, closely monitoring with ECG for myocardial ischemia is recommended. If cardiotoxicity appears, coronary artery disease should be ruled out. Cardiac computed tomography may be useful in this setting. Management in patients without significant coronary artery disease is controversial. Pretreatment with nitrates and/or calcium channel blockers may be considered, but drug rechallenge is usually recommended only when no other alternatives exist [20].

### 3.3. Antithrombotic Treatment in Cancer Patients

Antithrombotic treatment in cancer patients is always challenging because they present an increased risk of both thrombosis and bleeding events.

Venous thromboembolism (VTE) is a problem of utmost importance in cancer patients, which may be related to both the disease and its treatment. Diagnosis can be made in different environments, which include oncology and hematology clinics, internal medicine appointments and the emergency department. Historically, low molecular weight heparin (LMWH) has been the primary treatment for cancer patients with VTE and active cancer treatment. Currently, clinical trials have shown a good safety and efficacy profile of anti X factor anticoagulants to treat VTE in cancer patients [33,34,35,36,37]. The 2022 ESC Cardio-Oncology Guidelines recommend LMWH or Direct Oral Anticoagulants (DOACs) to treat VTE in cancer patients, both with an indication IA [17]. Anticoagulation is generally started by the attending physician during the acute phase. The long-term management and follow-up of these patients is often complex, so the expertise of a multidisciplinary team which includes hematologists and cardio-oncologists is key in achieving successful outcomes.

Regarding stroke prevention in cancer patients with atrial fibrillation (AF), there are not randomized clinical trials that have assessed DOACs in this specific population. However, subanalysis of the pivotal trials that evaluated the use of DOACs in non-valvular AF and many observational and real-life studies have shown that DOACs are effective and safe compared with warfarin. The Cardio-Oncology guidelines recommend the use of DOACs in non-valvular AF, the use of warfarin in valvular AF and restrict the use of LMWH to some specific clinical situations like severe thrombocytopenia, severe renal dysfunction or unoperated gastrointestinal or genitourinary tumors.

DOAC selection should be individualized based on thromboembolic risk, bleeding risk, comorbidities and drug–drug interactions. It is very useful to establish a close collaboration with the Pharmacy department and the Hematology department for the prescription and follow-up of DOAC treatment, as well as a multidisciplinary team approach. Lower dosage of DOACs than recommended is frequently observed in daily practice due to bleeding concerns, with a subsequent increase in thrombotic risk. Although evidence is scarce, the monitoring of DOACs plasma levels by an experienced hematologist in cancer patients receiving active therapy constitutes a reasonable strategy to assess the intensity of anticoagulation [38].

Bruton tyrosine kinase (BTK) inhibitors constitute a group of special interest. Its use has been associated to an increased risk of both AF and bleeding diathesis. Ibrutinib may increase warfarin levels via its effects on cytochrome P3A4 and raise dabigatran effects due to P-glycoprotein inhibition, so treatment with factor Xa inhibitors is recommended when anticoagulation is indicated. Due to the higher bleeding risk, BTK inhibitors should be temporarily interrupted in patients requiring dual antiplatelet therapy. Second-generation BTK inhibitors have shown a lower incidence of symptomatic cardiovascular events, but close monitoring is still recommended [17].

### 3.4. Myocarditis Related to Immune Checkpoint Inhibitors

Immune checkpoint inhibitors (ICIs), such as atezolizumab, durvalumab, ipilimumab, nivolumab and pembrolizumab, among others, are known to cause myocarditis. Although ICI-related myocarditis is uncommon (0.04–1.14%), it is associated with a high mortality (25–50%) and serious cardiovascular sequelae (46%) [39,40].

ICI-related myocarditis usually occurs early during treatment and is associated with an abnormal ECG and increased serum troponin levels. Approximately half of the cases present normal LV function. Cardiac magnetic resonance can be useful, as it will show myocardial oedema and late gadolinium enhancement. Endomyocardial biopsy still represents the gold standard for myocarditis diagnosis and should be considered if the diagnosis remains uncertain despite a comprehensive non-invasive approach. Sometimes, clinical diagnosis of ICI-related myocarditis can be challenging. Therefore, identification of concomitant immune-related adverse-events (irAEs) like myositis may help increase the suspicion. Recent studies have shown that the degree of troponin elevation is a predictor of adverse events [39].

A multidisciplinary approach to patients with ICI-related myocarditis is key. Severe cases of ICI-related myocarditis may be associated with neuro-muscular irAEs. As such, the evaluation and discussion of cases where myositis is present with a neurologist is essential. Moreover, myocarditis may be associated with other irAEs, which may require multidisciplinary evaluation with endocrinologists, pneumologists, dermatologists and gastroenterologists [41].

ICI-related myocarditis should be treated with high doses of steroids (methylprednisolone 1000 mg/day), as they are associated with lower troponin levels and better outcomes [39]. Therapy with ICIs should be withheld. Corticosteroids should be continued until symptom resolution and substantial improvement in troponin levels, LV function and conduction abnormalities. In those patients with steroid-refractory myocarditis or hemodynamic instability, other immunosuppressive therapies should be considered [42].

After clinical myocarditis, ICI therapy should be permanently discontinued. In the absence of an alternative therapy, the decision regarding restarting ICI therapy should be discussed in a multidisciplinary meeting [42].

### 3.5. Management of Patients with Cardiac Implantable Electronic Devices and Radiotherapy

Radiotherapy (RT) can affect cardiac implantable electronic devices (CIEDs), such as pacemakers and implantable cardioverter defibrillators (ICD). Occasionally, a device can interfere with RT delivery [43]. A multidisciplinary approach with radiation oncologists, cardio-oncologists and electrophysiologists is necessary to ensure the safety of patients with CIEDs receiving radiotherapy.

Before the beginning of a RT treatment, the following issues should be considered: (1) whether the device is a pacemaker or ICD; (2) whether the patient is pacemaker dependent; (3) the absorbed dose to the device; (4) the planned energy of the RT. The replacement of the device generator may be considered only when it is inside the therapeutic window and can affect adequate delivery of RT [43].

A complete device assessment before the beginning of RT and every few weeks during the delivery period should be carried out. Device reprogramming is also frequently required during RT.

Fradley M et al. have proposed an algorithm for these patients, classifying them as low or high-risk depending on their cumulative radiotherapy dose and neutron contamination, among others. Patients with >5 Gy cumulative radiotherapy dose, high neutron contamination, pacemaker dependent or those who have an ICD are considered high-risk and require close monitoring during radiation. Pulse oximetry registry and magnet application during RT sessions are recommended, as well as routine weekly device interrogations [32].

### 3.6. Pericardial Effusions

Pericardial disease is relatively common in cancer patients. It can be secondary to cardiac metastasis or can appear as a complication of cancer treatment. Malignant pericardial effusions are expected to develop in 5% to 15% of cancer patients with advanced disease [44].

Transthoracic echocardiography is the method of choice for the initial evaluation of patients with suspected pericardial effusion. It will be helpful not only in the diagnosis but also in the feasibility of pericardiocentesis [24]. Pericardial effusions should be quantified and graded to enable comparison in subsequent evaluations. Echocardiographic parameters of cardiac tamponade should always be analyzed.

Pericardial effusion treatment depends on the acuteness of symptoms, as well as the etiology. In pericardial effusion with no hemodynamic compromise, the treatment consists primarily of non-steroidal anti-inflammatory drugs and colchicine. Large pericardial effusions or those causing hemodynamic compromise may require pericardiocentesis [20]. During the working diagnosis of the primary cause of pericardial effusions, it is important to fully evaluate the pericardial fluid with cytology and flow cytometry. Sometimes, pericardial biopsy will be required. Recurrent malignant pericardial effusions should be treated with surgical pericardial window, pericardial sclerosis or balloon pericardiotomy [44].

In patients with radiation-induced cardiotoxicity, signs of constriction should be explored. Cardiac magnetic resonance can help at differentiating constrictive pericarditis from restrictive cardiomyopathy in cancer patients [24].

Figure 3 summarizes the different disease-specific clinical pathways.

### 3.7. Direct Cardiac Effects of Extracardiac Neoplasias

Some neoplasias have a direct or indirect effect on the heart. Primary amyloidosis (AL) is a systemic disease with multiorgan involvement associated with the deposit of an abnormal protein secondary to a plasma cell discrasia. The heart is affected in more than 50% of the patients and it is associated with poor prognosis. The treatment for AL amyloidosis is specific chemotherapy and stem cell transplantation, but the last one is inadvisable in patients with a significant heart disease. In some patients with advanced heart failure, heart transplant may be considered. Multidisciplinary teams comprising hematologists, cardiologists specialized in advanced heart failure and cardio-oncologists, among others, are mandatory for a proper approach to this disease [45].

Carcinoid syndrome is caused by neuroendocrine tumors, which secrete vasoactive substances that may affect gastrointestinal motility, produce bronchospasm, flushing or hypotension. The term carcinoid heart disease reflects heart involvement, which occurs in 50–70% of patients, and mainly consists of tricuspid and pulmonary valves stenosis and regurgitation. Carcinoid syndrome medical treatment is based on somatostatin analogues. Patients with carcinoid heart disease may also need heart failure specific treatment, as well as cardiac surgery. Valve replacement is generally preferred over repair. As carcinoid tumors may affect different organs, a multidisciplinary team that includes oncologists, endocrinologists, cardiac surgeons, nuclear medicine and cardio-oncologists is imperative [46].

## 4. Future Directions

Cardio-Oncology is a growing and evolving field, so there are still quite unmet needs. First, Cardio-Oncology units should expand and become established in every single hospital that treats cancer patients. A dedicated training core curriculum and formal recommendations to create programs of excellence are required. Additionally, artificial intelligence will play a key role in pretreatment risk assessment. Modern biomarkers and new technologies will allow earlier and more precise detection of cardiotoxicity. Big data based on large Cardio-Oncology registries will generate evidence and help the decision-making. Finally, social determinants of health, patient reported outcome measures and patient reported experience will be a driving force in deciding the most appropriate individual management.

## 5. Conclusions

Cardio-Oncology now goes far beyond ventricular dysfunction screening and treatment. Cardiotoxicity comprises a wide spectrum of myocardial, pericardial, coronary and arrhythmic complications. Cardio-Oncology units should perform thorough cardiovascular management of cancer patients to enhance their quality of life, avoid unnecessary withdrawal from cancer therapies and improve their prognosis. This is a complex and challenging job that requires a multidisciplinary approach and continuous communication among all parties involved, including the patients. A comprehensive and structured program is the key to success. The Cardio-Oncology team should be patient-centered, but should also focus on research, training and innovation.

## Figures and Tables

**Figure 1 cancers-15-05885-f001:**
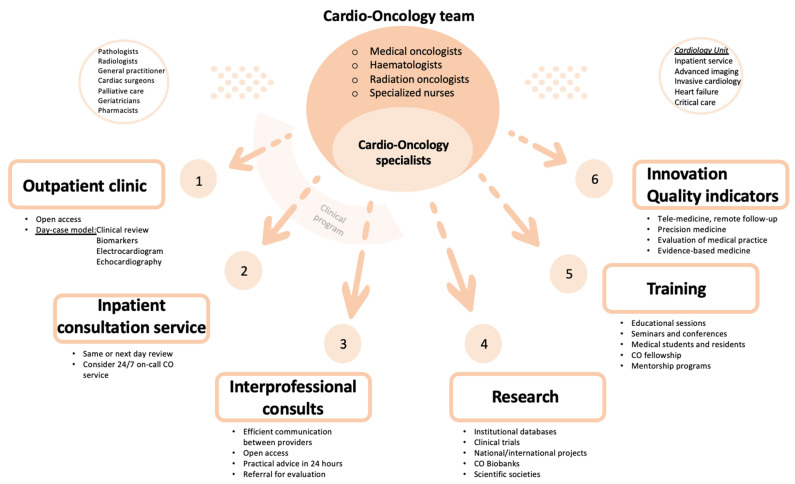
Main components of a Cardio-Oncology Program. Figure shows the different areas a Cardio-Oncology program should cover and its main components. Abbreviations: CO, Cardio-Oncology.

**Figure 2 cancers-15-05885-f002:**
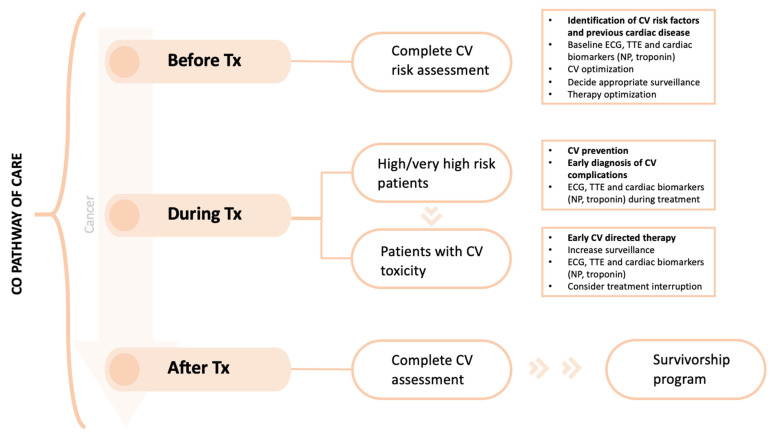
Cardio-Oncology clinical care pathway. Figure shows the management of cancer patients before, during and after cancer therapy. Abbreviations: CO, Cardio-Oncology; CV, cardiovascular; ECG, electrocardiogram; NP, natriuretic peptides; TTE, transthoracic echocardiography; Tx, treatment.

**Figure 3 cancers-15-05885-f003:**
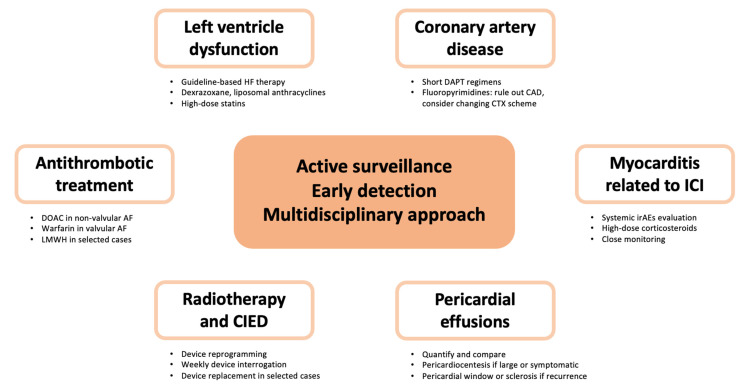
Disease-specific clinical pathways. Figure shows the main measures to take into account when a specific cancer-related cardiovascular complication appears. Abbreviations: AF, atrial fibrillation; CAD, coronary artery disease; CIED, cardiac implantable electronic devices; CTX, chemotherapy; DAPT, dual antiplatelet treatment; DOAC, direct oral anticoagulants; ICI, immune checkpoint inhibitors; irAEs, immune-related adverse events; LMWH, low-molecular-weight heparin.

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
