# Peer review of "Developing Cardio-Oncology Programs in the New Era: Beyond Ventricular Dysfunction Due to Cancer Treatments"

_cancers, 2023, doi:10.3390/cancers15245885_

Round 1

Reviewer 1 Report

Comments and Suggestions for Authors

The review is well organized and the illustrations help  the reader in this rather complex field.

I have just a concern/suggestion:

In may opinion the problem of venous thromboembolism (VTE) deserves more attention. by cardio-oncologists who should not simply delegate the problem to haemostasiologists. As known, VTE is differently managed in different countries involving internists, angiologists, haematologists, cardiologists etc depending on tradition and on health care organization. How is this dealed with in Spain? What are suggestions of authors. I suggest to dedicate a paragraph on this point

Comments on the Quality of English Language

good quality

A typo at page 5, line 168

"farmacysts" should read "pharmacists"

Reviewer 2 Report

Comments and Suggestions for Authors Cardiovascular toxicities of anti-cancer drugs remain a major concern, though modern anti-cancer medications have achieved a significant improvement. A specific Cardio-Oncology Unit is pivotal to allow patients with cancer to receive the best treatment approach while minimizing adverse cardiac effects. The development of structured Cardio-Oncology programs contributes to better patients’ medical care. In this review, the authors summarized their experience and described the essential steps to consider when creating a Cardio-Oncology program. This manuscript is significant for an improved medical care of the patients with cancers, hence is interesting. Comments on the Quality of English Language

The quality of English language is good.
